# A whole-ecosystem experiment reveals flow-induced shifts in a stream community

Daniela Rosero-López [1,7 ✉], M. Todd Walter[1], Alexander S. Flecker[2], Bert De Bièvre[3], Rafael Osorio[4], Dunia González-Zeas[5], Sophie Cauvy-Fraunié[6] & Olivier Dangles [5]

The growing threat of abrupt and irreversible changes to the functioning of freshwater ecosystems compels robust measures of tipping point thresholds. To determine benthic cyanobacteria regime shifts in a potable water supply system in the tropical Andes, we conducted a whole ecosystem-scale experiment in which we systematically diverted 20 to 90% of streamflow and measured ecological responses. Benthic cyanobacteria greatly increased with a 60% flow reduction and this tipping point was related to water temperature and nitrate concentration increases, both known to boost algal productivity. We supplemented our experiment with a regional survey collecting > 1450 flow-benthic algal measurements at streams varying in water abstraction levels. We confirmed the tipping point flow value, albeit at a slightly lower threshold (40-50%). A global literature review broadly confirmed our results with a mean tipping point at 58% of flow reduction. Our study provides robust in situ demonstrations of regime shift thresholds in running waters with potentially strong implications for environmental flows management.

[1] Soil and Water Lab, Department of Biological and Environmental Engineering, Cornell University, Ithaca, NY, USA. [2] Department of Ecology and Evolutionary Biology, Cornell University, Ithaca, NY, USA. [3] Fondo para la Protección del Agua FONAG, Quito, Ecuador. [4] Gerencia de Ambiente e Hidrología, Empresa Pública de Agua Potable y Saneamiento EPMAPS, Quito, Ecuador. [5] Université de Montpellier, Centre d'Ecologie Fonctionnelle et Evolutive, UMR 5175, CNRS, Université Paul Valéry Montpellier, EPHE, IRD, Montpellier, France. [6] INRAE, UR RIVERLY, Centre de Lyon-Villeurbanne, Villeurbanne, Cedex, France. [7] Present address: Universidad San Francisco de Quito USFQ, Instituto Biósfera, Laboratorio de Ecología Acuática, Calle Diego de Robles y Pampite, Quito, Ecuador. ✉email: dr527@cornell.edu

Understanding ecosystem responses to environmental disturbances is crucial for making robust ecological predictions in an increasingly stressed world[1,2]. While some ecosystems respond gradually, others undergo sudden shifts in structure and function after disturbance exceeds a critical threshold[3]. Such shifts can result in transitions to ecosystem states for which recovery could be slow (e.g., the system displays hysteresis) or impossible (the system moves to an alternative stable state)[4,5]. Lake-trophic-state is one of the best-studied examples of ecological transitions, where clear waters have been documented to suddenly change to algal-dominated conditions (i.e., nutrient loading, low water level)[2,5–7]. More generally shifts in aquatic ecosystem states have dramatic consequences for biodiversity, for water provisioning, and other vital ecosystem services for humans (i.e., fisheries, agriculture, aesthetics)[8,9]. Therefore, the study of ecosystem state shifts is crucial to developing sustainable management plans for increasingly threatened natural resources.

Our detailed comprehension of ecosystem state shifts in lakes has been possible through ecosystem-scale experiments[1,2,10,11]. In comparison, there are less data regarding regime shifts in natural running waters at ecosystem-scales because flow manipulations of free-flowing rivers are logistically challenging, currently limited to experimental releases from existing hydraulic structures[12–16]. Consequently, our understanding of flow-biota relationships in running waters has mostly relied on correlations using field data from largely uncontrolled stream reaches or data from experiments performed in artificial channels and flumes[7,12,17]. Therefore, the sustainable management of running waters is currently based on limited information to detect critical flow-ecology thresholds and establish the quantity and timing of water necessary to balance ecological and human needs of river systems, i.e., environmental flows (e-flows)[13,17,18,19]. The understanding of ecosystem state shifts in running waters could provide early-warning signals for managing water resources at watershed scales that, in mountain systems, often include lakes and reservoirs that could be manipulated. Indeed, streams have relatively small instantaneous water volume when compared to lakes and may therefore respond more quickly to environmental changes[20] (even though the continuous water renewal may also increase a stream's resistance to stress).

Defining flow-induced shifts in running waters appears particularly crucial in the case of benthic cyanobacteria harmful algal blooms (CyanoHABs), where several taxa (e.g., *Anabaena*, *Oscillatoria*, *Lyngbya*, *Nodularia*) produce toxins with potentially severe effects on human, livestock, pets, and wildlife health[2,20–23]. Benthic cyanoHABs are a global phenomenon, whose emergence in running freshwaters is often triggered by nutrient loading and decreases in water levels with associated increases in temperature[7,13,22,24]. Benthic cyanoHABs recurrence in rivers also imperils lakes and reservoirs in potable water systems and vice versa (Fig. 1a). Benthic cyanoHABs are expected to become more frequent and more impactful due to global warming, widespread increases in water use, and nutrient inputs[8,9,13]. While planktonic CyanoHABs have been well addressed in lakes and large impounded rivers (e.g., 3, 11), our knowledge is much more limited in benthic cyanobacteria of running waters, where ecological drivers (e.g., temperature, pH, dissolved oxygen) of benthic cyanoHABs and dynamics of system transitions are poorly understood (see refs. 7,20,25,26).

Here, we address regime shifts in benthic cyanoHABs and other benthic organisms (e.g., diatoms, green algae, and grazing invertebrates) using a whole ecosystem experiment in a mountain river system of the tropical Andes (Fig. 1a–c). The tropical Andes are a highly biodiverse region impacted by the rapid proliferation of water infrastructure and at the same time provides potable water to main cities[27,28]. Using a Before-After-Control-Impacted

(BACI) design[29,30] and a series of weirs[31], we gradually reduced and reinstated the flow of a downstream experimental reach by diverting 20 to 90% of the upstream flow (manipulated stream, see Fig. 1a–c) while a similar adjacent reference stream remained un-manipulated (control stream). The stream analysis consisted of three phases[1]: establishment of baseline conditions (BL) under unaltered flow (~1.5 years)[2]; experimental diversion of flow, inducing systematic flow reductions (FR) in the downstream reach (~0.3 years), and[3] gradual reset to initial flow (FI) conditions in the downstream reach (~0.3 years) (Fig. S1). This experimental design allowed us to (1) assess the response of benthic communities to flow reduction, with a special focus on cyanobacteria biomass and quantify potential flow thresholds that trigger system shifts and (2) evaluate our study system's resilience, defined as the capacity of the benthic communities to return to their initial configuration after flow disturbance ceases. Additionally, we compared benthic cyanobacteria biomass between sites upstream and downstream of water intakes in seven streams (21 stream-sites) sampled weekly (~2 years: 1456 paired data) (see Methods). These dual approaches enabled us to evaluate whether experimental results were consistent with the benthic cyanobacteria response to permanent flow alteration across a gradient of flow reduction in the same study area (from 98% to 23%). Also, we performed a literature review of qualitative and quantitative benthic cyanobacteria biomass response to changes in flow levels (see Fig. 1d). Comparisons among these studies could help predict future temporal ecological variability under reductions of flows caused by water abstractions and potential exacerbations by climate change, especially in naturally nutrient-poor ecosystems that supply water for humans [32].

## Results

**Whole-ecosystem experiment.** Except for a spate event at day 90, our experimental design was successful in gradually reducing and then increasing flows in the manipulated stream reach compared to the baseline conditions and those in the un-manipulated control stream (upstream and downstream sites) (Fig. 2a, b and Table 1). The experimental flow manipulation had no significant effect on stream physical and chemical characteristics (Table 1 and Fig. S2), except for water temperature and nitrate concentrations with variations through time inversely correlated to variations in flow (Fig. 2c, e).

A multivariate, autoregressive state-space analysis revealed a significant positive effect of flow on benthic cyanobacteria biomass related with the effect on temperature and nitrate concentration increase ($C_{Flow \to Cyano} = 0.031$, $C_{Temp \to Cyano} = 0.016$, $C_{NO_3^- \to Cyano} = 0.023$, $AIC_c = 161.1$; see Methods section). We found no significant inter-specific interaction with benthic algae and invertebrates. The state-space residual analysis determined two significant shifts in cyanobacteria biomass during the experimental manipulation: benthic cyanobacteria biomass shifted to high levels (+173% of initial values) after a 60% flow reduction and returned to initial biomass levels when 60% of upstream flow was restored (~6 weeks) (Fig. 3a, red line). The experimental flow reduction favored cyanobacteria dominance over other types of benthic algae that also changed (i. e., diatoms increased, and green algae remained stable) but the proportion of the three types returned to baseline conditions after flow was reinstated (Fig. S3a). The experimental flow manipulation had no significant effect on benthic invertebrates' biomass (BI) even though biomass tended to decrease during flow reduction compared to baseline conditions ($BI_{FR} = 3.63$ g DM. m$^{-2}$ ± 1.02, $BI_{BL} = 3.94$ g DM. m$^{-2}$ ± 0.79) (Table 1 and Fig. S2a). We found no significant shifts in benthic cyanobacteria, algal, and invertebrate biomass on the upstream and downstream reaches of the control stream (Figs. S2a and S3b).

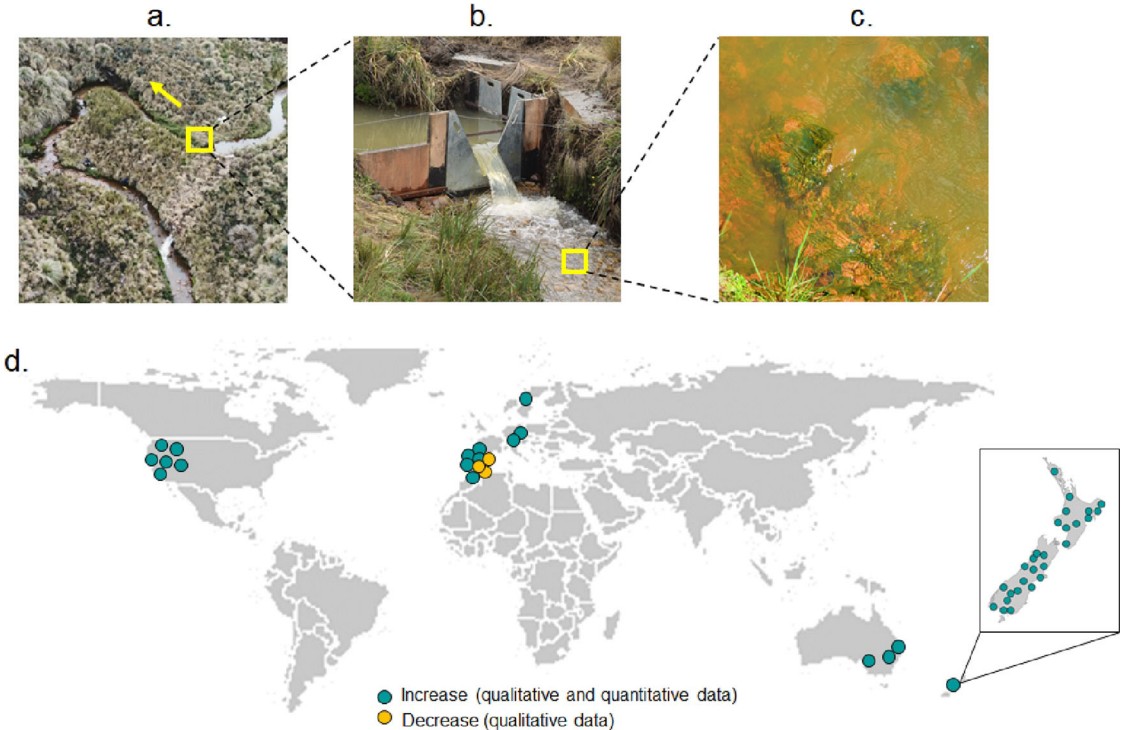

**Fig. 1 Benthic cyanobacteria relations to flow in rivers. a** Experimental flow manipulations in a free-flowing high-altitude stream reach, **b** using a series of v-notch weirs to reduce natural flow in fixed percentages, **c** showed an increase in benthic cyanobacteria levels, **d** worldwide studies (circles, $n = 53$) reporting qualitative observations of benthic cyanobacteria biomass decreases (yellow) and qualitative and quantitative observations of biomass increases (teal-blue) with respect to flow reduction. Right-downside frame show data reported for New Zealand.

**Regional and global scale surveys**. Our experimental results were supported by a complementary field survey of seven nearby streams covering a gradient in flow reduction due to water abstraction activities (between 98% to 23%). We found a significant benthic cyanobacteria biomass shift threshold when 40% of the upstream flow was abstracted ($R^2 = 0.57$, $p = 0.021$), showing a breakpoint ($\varepsilon = 0.6$) on the relation according to the Rammer–Douglas–Peucker (RDP) model (Fig. 4a, green line). Further, our global survey of flow-benthic cyanobacteria biomass relationships in running water systems (33 sites (Fig. 1d) and, Table S1) revealed that when at least 50% of the baseline flows were preserved, there was no effect on benthic cyanobacteria biomass. After passing this threshold some systems dramatically increased their cyanobacteria biomass (up to a four-fold increase when less than 10% of the flow remained in the stream) while others showed low increase at the same levels of flow reduction (Fig. 4b, blue line). The RDP model fitted to these data indicated a benthic algal biomass breakpoint at a 58% flow reduction for a significant correlation between cyanobacteria increase factor and the relative percentage of flow reduction ($R^2 = 0.79$, $p = 0.032$) (Fig. 4b, teal-blue line).

## Discussion

Previous studies on ecological response to reduced flow in running waters have commonly found significant changes in benthic community structure and function characterized by a change in biomass and density[12,16,26,33,34], reduction in taxon richness[14,18], and alteration of the trophic organization[14]. The impacts, however, vary among study systems (e.g., natural flow regime vs. water quality vs. the features of the disturbance imposed)[35–37]. Low flows decrease water velocities, which favor benthic algal attachment onto the substrate[16,38,39] and generate higher water temperatures that boost primary production[24,40]. Our experimental manipulation confirmed this general pattern, even though

it was performed in oligotrophic streams showing relatively low temperatures and nutrient poor concentrations under natural conditions. These temperature and nutrient increases with flow reduction confirm that mountain streams are sensitive to water abstraction and could abruptly shift to an alternate state when flows are altered[14,27].

Our study provides strong evidence that lotic systems may undergo sudden shifts in structure and function of benthic communities after flow reduction exceeds a critical threshold. The combination of experimental manipulation, field monitoring, and a global literature survey suggests that a flow reduction beyond a threshold of about 40-60% of natural flow conditions induces abrupt shifts in cyanobacteria biomass. Cyanobacteria is of particular interest as its increase/dominance in the benthic community jeopardizes the quality of potable water. A 40–60% flow reduction coincides with thresholds found for other organisms (e.g., invertebrates)[41,42] and is well above the legal environmental flow thresholds recommended in many countries (e.g., from the mean annual flow: 10% in Ecuador, 25% in Colombia, and 20% in Chile). Our findings, therefore, call for a revision of tipping points appointed in current water frameworks to ensure sustainable water management at the watershed scale. We recommend considering benthic cyanobacteria thresholds as they can be scoured and moved down streams and rivers threatening lakes and reservoirs[22,43,44], interconnected in potable water systems[45,28].

Beyond the quantification of a flow reduction threshold inducing benthic cyanobacteria biomass -shift, our experiment also allowed identification of the ecosystem capacity to recover from gradual low flows. We found that benthic cyanobacteria biomass would return to pre-manipulated values when flow reduction is about 40% of the baseline flow, 20% more than the threshold that triggers benthic algal proliferation during the drying phase. As a potential side effect, strong precipitation during our flow recovery stage might have led to an uncontrolled

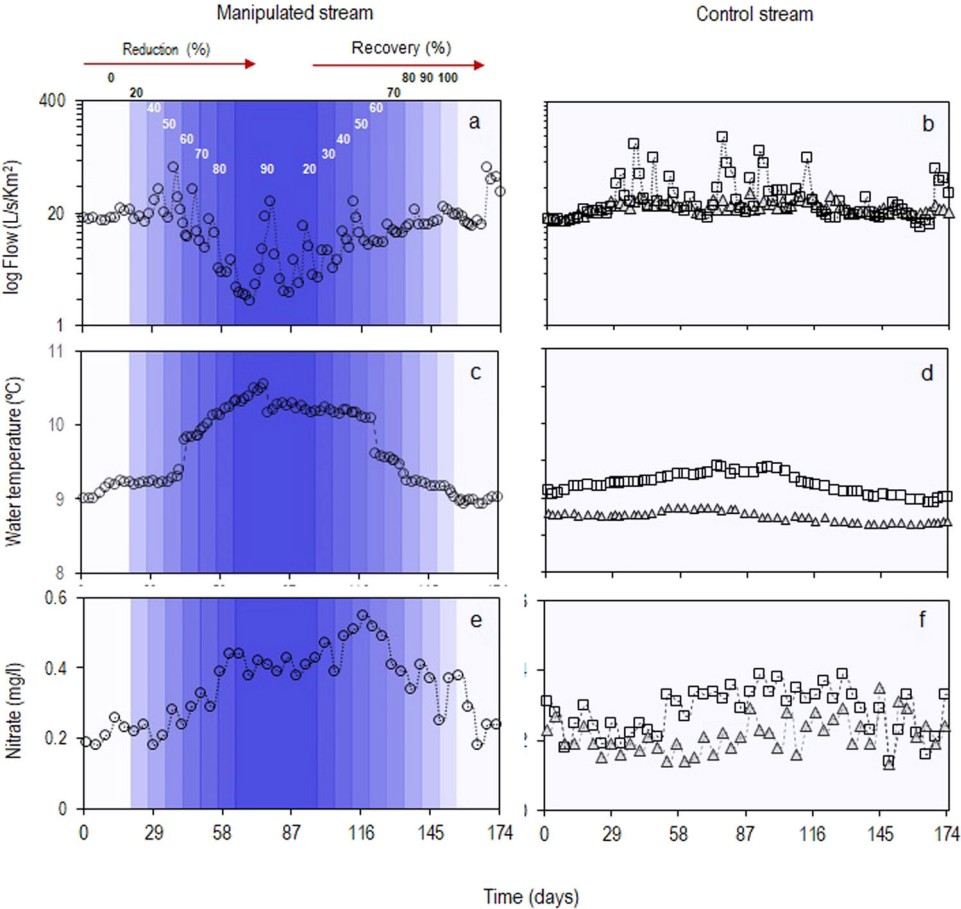

**Fig. 2 Environmental variables during experimental flow manipulations and natural flow. a**, **c**, **e** Time-series (open symbols) and smoothed-state estimates (dotted lines) for the manipulated stream (circles) during gradual flow reductions and recoveries of the 20, 40, 50, 60, 70, 80, 90%, and 20, 30, 40, 50, 60, 70, 80, 90, and 100% of upstream flow (blue shades), while maintaining daily fluctuation (**b**, **d**, **f**) natural flow conditions for the upstream reach of the experimental site (squares) and the upstream reach of the control site (triangles).

flow increase and may have facilitated rapid cyanobacteria bio-mass return to natural levels, in contrast to that observed in other systems[2,20]. The observed symmetrical response of cyanobacteria biomass recovery to flow increase contrasts with other studies reporting longer recovery times of benthic communities, despite the quicker turnover[14]. This may be explained by the relatively short duration of low flows in our experiment, which impeded the organization of benthic food webs, as suggested by the non-significant effect of flow reduction on grazing invertebrates and the small tendency in invertebrate decrease during flow recovery[12–14]. It is likely that more prolonged flow reductions would prevent the system from fully recovering between con-secutive flow disturbances, with potentially severe consequences on water quality indicated by the nitrate concentration increase[8,21,44]. A longer time flow reduction might elucidate the underlying mechanism for nitrate increase as we observed dia-toms had a delayed response while green algae showed none. Such pressures may also have environmental consequences beyond the threshold's timeline, with underestimated effects on the system[46].

Complementary studies testing various scenarios of flow alteration - gradual in intensity, constant in duration, and fre-quency - are needed to fully understand the interplay between these three dimensions of hydrological stress to fully apprehend freshwater resilience to algal proliferation[16,19,20,33,47]. However, by combining experimental manipulations, field surveys, and literature analysis in this work calls for more integrative

approaches to defining environmental flows adapted to specific systems. As flow alterations are among the most pressing threats to the integrity of stream ecosystem functioning and the persis-tence of freshwater species, managers should be particularly attentive to developing sound guidelines to help achieve sus-tainable management of river flows.

## Methods

**Study area**. The study was conducted in the headwaters of the Chalpi Grande River watershed, 95 km[2], located inside the Cayambe-Coca National Park in the northern Andes of Ecuador at an elevation range of 3789 to 3835 m (S 0°16′ 45″, W 78° 4′49″). This watershed harbors the primary water supply system for Quito. The system includes two reservoirs and 10 water intakes placed on first and second-order streams that, altogether, provide 39% of Quito's water supply[28]. We mon-itored the Chalpi Norte stream for ~1.5 years prior to conducting our experiment for ~0.5 years (176 days), and ~0.4 years after the manipulation. Further, in the nearby area, we monitored 21 stream sites distributed upstream and downstream water intakes from the supply system (Fig. S4).

**Experiment for flow manipulation and monitoring flow reduction and recov-ery**. We conducted our experimental flow manipulation between October 2018 and April 2019 in a mainly rain-fed stream[45]. The experiment manipulated natural flows encompassing stable low flows and sporadic spates characterizing the high temporal variability of headwaters[45,28] (Figs. 2a, b and S1). We set up a full Before-After/Control- Impact (BACI) experiment[29] to evaluate ecosystem variables under natural and manipulated flow conditions. We identified a free-flowing stream reach on the Chalpi Norte that was above any water intakes that allowed us to divert flow with an ecohydraulic structure[31]. The structure was located above a meander, which we used to divert flow and return it to the stream below the meander (Fig. S4). The experimental site was comprised of an upstream/free-flowing reach

**Table 1 Mean abiotic and biotic variables during the experimental phases (BL, FR, FI re-instate as flow recovery) in four experimental reaches in two streams (manipulated and control).**

| | Manipulated stream | | | | | | Control stream | | | | | |
| | Upstream site | | | Downstream site | | | Upstream site | | | Downstream site | | |
| | BL | FR | FI | BL | FR | FI | BL | FR | FI | BL | FR | FI |
|---|---|---|---|---|---|---|---|---|---|---|---|---|
| Discharge (m³ s⁻¹) | 0.188 | 0.211 | 0.159 | 0.188 | 0.087ᵃ | 0.103ᵃ | 0.019 | 0.021 | 0.023 | 0.019 | 0.022 | 0.024 |
| (% CV) | 57 | 82 | 69 | 57 | 99 | 91 | 19 | 30 | 24 | 18 | 29 | 22 |
| Temperature (°C) | 7.4 | 7.7 | 8.1 | 7.4 | 10.1ᵃ | 9.5ᵃ | 6.9 | 6.9 | 7.1 | 7.1 | 6.9 | 7.1 |
| (min–max) | 4.1–10.5 | 4.7–10.6 | 5.1–11.1 | 4.3–10.5 | 5.6–11.7 | 5.1–11.1 | 4.1–9.4 | 4.1–9.3 | 4.2–9.6 | 4.2–9.5 | 4.1–9.5 | 4.1–9.3 |
| pH | 6.9 | 6.8 | 6.8 | 6.9 | 7.2 | 6.8 | 6.8 | 6.8 | 7.0 | 6.8 | 6.9 | 6.9 |
| (±SD) | 0.3 | 0.3 | 0.4 | 0.5 | 0.6 | 0.4 | 0.3 | 0.4 | 0.3 | 0.4 | 0.3 | 0.2 |
| Conductivity (µS cm⁻¹) | 40.03 | 44.05 | 42.01 | 44.77 | 48.23 | 43.07 | 38.55 | 38.39 | 41.01 | 39.74 | 37.88 | 40.71 |
| (±SD) | 2.33 | 3.21 | 2.81 | 2.88 | 4.91 | 3.71 | 1.55 | 2.11 | 1.65 | 1.56 | 2.11 | 1.65 |
| Dissolved oxygen (mgL⁻¹) | 7.91 | 7.75 | 8.24 | 7.92 | 7.69 | 7.88 | 8.12 | 7.89 | 7.73 | 7.79 | 7.99 | 7.54 |
| (min–max) | 6.1–9.9 | 6.9–9.7 | 6.2–9.9 | 6.6–10.3 | 6.2–9.2 | 6.1–9.4 | 6.3–9.4 | 6.5–9.9 | 6.4–10.1 | 6.3–9.8 | 6.5–9.9 | 6.5–10.1 |
| Nitrate (mg L⁻¹) | 0.34 | 0.24 | 0.34 | 0.26 | 0.36ᵃ | 0.48ᵃ | 0.16 | 0.17 | 0.18 | 0.16 | 0.16 | 0.18 |
| (±SD) | 0.04 | 0.07 | 0.02 | 0.06 | 0.13 | 0.05 | 0.05 | 0.03 | 0.09 | 0.06 | 0.03 | 0.09 |
| Phosphate (µg L⁻¹) | 0.07 | 0.07 | 0.09 | 0.09 | 0.08 | 0.08 | 0.05 | 0.06 | 0.05 | 0.07 | 0.07 | 0.08 |
| (±SD) | 0.004 | 0.005 | 0.003 | 0.005 | 0.005 | 0.006 | 0.002 | 0.002 | 0.004 | 0.005 | 0.003 | 0.006 |
| Cyanobacteria (µg Chl-a cm⁻²) | 1.86 | 1.89 | 1.63 | 1.22 | 3.95ᵃ | 3.23ᵃ | 0.99 | 1.04 | 1.08 | 0.95 | 0.97 | 1.05 |
| (±SD) | 0.39 | 0.39 | 0.32 | 0.45 | 2.12 | 2.25 | 0.07 | 0.21 | 0.22 | 0.09 | 0.19 | 0.17 |
| Diatoms (µg Chl-a. cm⁻²) | 2.12 | 2.04 | 1.97 | 1.92 | 2.67ᵃ | 2.86ᵃ | 1.48 | 1.58 | 1.45 | 1.32 | 1.62 | 1.52 |
| (±SD) | 0.36 | 0.43 | 0.38 | 0.53 | 1.04 | 1.09 | 0.29 | 0.34 | 0.34 | 0.14 | 0.32 | 0.31 |
| Green algae (µg Chl-a cm⁻²) | 2.08 | 1.43 | 1.64 | 1.95 | 1.18 | 0.94 | 1.26 | 1.07 | 1.01 | 1.24 | 0.95 | 0.91 |
| (±SD) | 0.33 | 0.39 | 0.40 | 0.43 | 0.48 | 0.54 | 0.13 | 0.44 | 0.34 | 0.15 | 0.18 | 0.21 |
| Invertebrates (g DM m⁻²) | 3.85 | 4.01 | 4.61 | 3.94 | 3.63 | 4.11 | 2.99 | 2.34 | 2.94 | 2.83 | 3.02 | 2.54 |
| (±SD) | 0.51 | 0.69 | 0.97 | 0.79 | 1.02 | 0.89 | 0.54 | 0.43 | 0.78 | 0.48 | 0.64 | 0.73 |

*BL* baseline, *FR* flow reduction, *FI* flow.
ᵃIndicates significant differences in parameter value when compared to values from the baseline conditions before alteration (significant *p* values at 0.05 alpha level from a paired one-tail *t*-test).

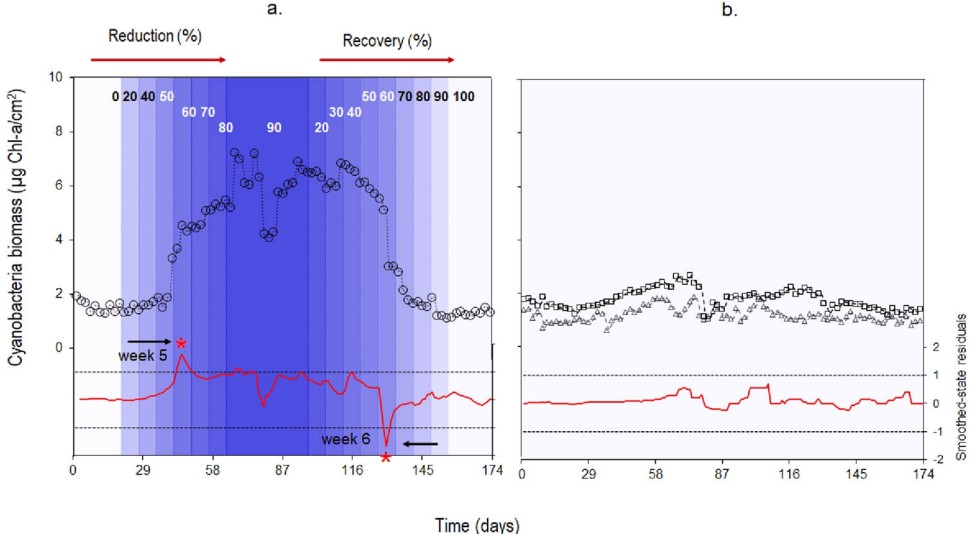

**Fig. 3 Benthic cyanobacteria biomass response to experimental manipulations and natural flow. a** Time-series and smoothed-state estimates (dotted lines) for the experimental site on the manipulated stream (open circles) undergoing reductions and recovery of flow (blue shades); **b** time-series of the upstream reach of the experimental site (squares) and the upstream reach of the control site (triangles) under natural flow conditions. Red curves in lower scale correspond to standardized smoothed-state residuals from state-space models for the experimental site and the upstream reach of the control site (the downstream reach shows a similar response to the upstream site as there is no alteration in the stream, see Table 1); dashed black lines are the upper and lower 95% confidence intervals, and stars indicate when standardized smoothed-state residuals are beyond the dashed line of confidence interval levels.

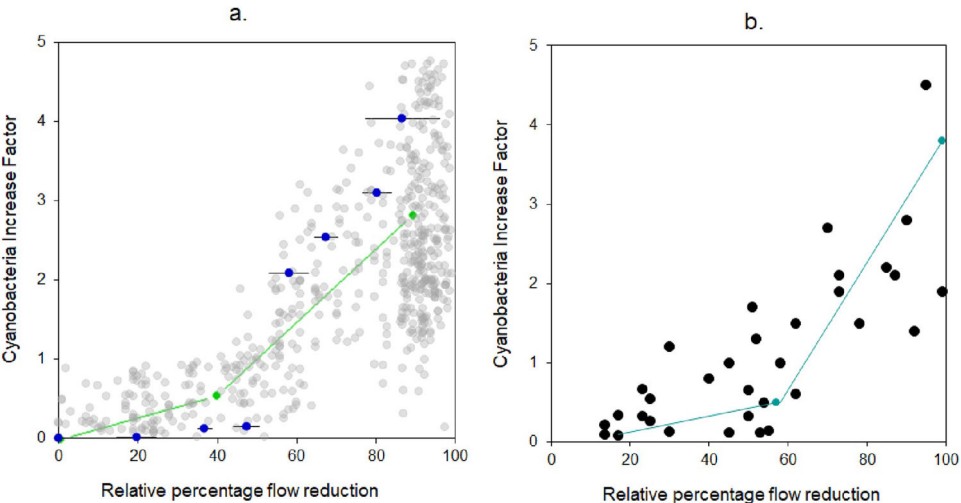

**Fig. 4 Benthic cyanobacteria increase factor according to the relative percentage of flow reduction. a** Measurements (gray circles) of paired cyanobacteria-flow monitoring data ($n = 697$) at the time of the sample in one location upstream of the water intake and two locations downstream water intakes in seven streams from the water supply system; experimental results (blue circles) of cyanobacteria increase with targeted flow reductions including variations ±SE from temporal replicates within flow reduction (black lines). Rammer–Douglas–Peucker model (RDP) (green line) fitted to monitoring data showing a breakpoint for cyanobacteria increase with a 40% flow reduction. **b** A global survey of benthic cyanobacteria-flow data ($n = 33$) showing a distribution fitted with RDP model: cyanobacteria increase after a breakpoint of 58% flow reduction (teal blue line).

($L = 25$ m) (reference conditions), located ~32 m above the ecohydraulic structure and a downstream/regulated reach ($L = 97$ m) located immediately below the flow manipulation structure (Fig. 1b–d)[31]. The control site was located in a free-flowing stream, a tributary of the Chalpi Norte stream, with an upstream reach separated from a downstream reach by a distance of 16 m. We manipulated the instantaneous flow of the Chalpi Norte stream through a series of fixed percentages using different v-notch weir pairs[31]. We started diversions to maintain in the meander 100, 80, 60, 50, 40, 30, and 20% of the incoming flow for 7-day periods (based on local observations of benthic algal colonization); then we maintained 10% of the upstream flow for 36 days. We started to return flow gradually to recover 20, 30, 40, 50, 60, 80, and 100% of the upstream flow. In response to a natural spate while we maintained the 10% of upstream flow, the manipulated flow briefly (during ~9 h)

increased above the targeted reduction (i.e., 54% instead of 10%) (Fig. 2a). We registered the spate of flow on the upstream reach of the experimental site (Figs. 2b and S1).

**Stream monitoring in adjacent streams**. We monitored 21 stream sites between July 2017 and July 2019. We selected seven streams with water intakes placed on the main channel (Chalpi Norte, Gonzalito, Quillugsha 1, 2, 3, Venado, and Guaytaloma). We sampled one site upstream of the water intake and two sites (i.e., 10 m and 500 m) downstream to obtain a wide range of flow reduction levels (Fig. S4) (see, 30 for further details on stream sites).

**Global literature survey**. We performed a systematic literature review to explore benthic algae responses to flow alterations (increase or decrease), focusing on cyanobacteria in streams. We used ISI Web of Science, Google Scholar, and Google Search for the entries: "benthic cyanobacteria" + "stream", and "river", "benthic algal bloom" + "flow" and all available combinations (Table S1). We selected papers containing information on benthic cyanobacteria and algae biomass and flow or water level measurements; specifically, we explored detailed information regarding experiments, spatial studies with upstream and downstream sites, and temporal replicates, as well seasonal associated benthic cyanobacteria blooms. We used published and/or publicly available data to calculate the percent of flow alteration in streams and calculated a factor for cyanobacteria biomass increase or decrease (quantitative studies) according to reported baseline conditions (either temporal or spatial). Only three out of 53 study sites reported a qualitative decrease in benthic cyanobacteria biomass attributable to flow reduction (Fig. 1d). Most studies (94%, $n = 50$) reported biomass increases with flow reductions. Among these studies sites, 44% reported qualitative observations where low flows were proposed as one of the environmental drivers responsible for benthic cyanobacteria blooms. While 66% of study sites ($n = 33$) related cyanobacterium biomass increase in time or space due to flow reductions caused by droughts, extreme low flow events, water abstractions, and experimental flumes manipulations.

**Abiotic and biotic variables sampling and analyses**. Water level sensors recording every 30 min (HOBO U40L, Onset USA) were installed at both upstream and downstream sites of water intakes, and on the experimental and control stream reaches (BACI desing), where we conducted multiple wading-rod flow measurements to convert water level into discharge via stage-discharge relationships (ADC current meter, OTT Hydromet, Germany). Streamwater's physical and chemical in situ parameters (i.e., pH, temperature, conductivity, dissolved oxygen) were measured three times during biotic sampling on both stream sites and adjacent streams using a portable sonde (YSI, Xylem, USA). We collected water samples (500 ml) during in situ samplings to analyze nutrients (i.e., nitrate and phosphate) at the water supply company's (EPMAPS) laboratory. We also measured precipitation from a rain gauge (HOBO Onset USA) installed in the Chalpi Norte stream.

Our biotic variables included three benthic algae: cyanobacteria, diatoms, and green algae), and aquatic invertebrates biomass (Table 1). To measure Chl-a from cyanobacteria and benthic algae on artificial substrates, we used a BenthoTorch® (bbe Moldaenke GmbH, Germany) on unglazed ceramic plates (200 mm × 400 mm) with a grid of 25 squares of 2500 mm² to allow algal accrual on a standardized surface. We allowed 21 days for colonization (based on previous observations) and then we placed all substrates[5] at the beginning of the experiment. We performed five readings on five squares randomly selected within each plate. To consider the effect of benthic invertebrates to flow variations, we sampled stream sites using a Surber net (mesh size = 250 μm, area = 0.0625 m²). On the experimental and control sites we measured biotic, physical, and chemical in situ parameters every two days ($n = 1760$), and nutrients and invertebrates every seven days ($n = 500$) for the duration of the flow manipulation (~0.5 years). On the monitored sites, we measured biotic, physical, and chemical in situ parameters every seven days ($n = 1456$) and nutrients and invertebrates every 30 days ($n = 336$). To evaluate differences we calculated mean abiotic and biotic variables during the different phases (BL: baseline, FR: flow reduction, FI: gradual reset to initial flow) in the four-stream reaches to apply the BACI design[29]: upstream and downstream reaches on the experimental and control sites. We applied a paired one-tail $t$-test at $\alpha = 0.05$ to compare FR and FI phases to baseline conditions, based on the expected direction of the response [1,14].

**Statistics and reproducibility**. To quantify the relationships between environmental variables and cyanobacteria biomass under manipulated and natural flow conditions, including interaction among algae and with invertebrates, we used multivariate autoregressive state-space modeling (MARSS)[14,30]. We fitted models with Gaussian errors for flow, conductivity, pH, water temperature, nitrate, phosphate, cyanobacteria, benthic algae, and invertebrate biomass time series via maximum likelihood (MARSS R-package)[48]. The state processes $X_t$ includes state measurements for all four benthic components (cyanobacteria, diatoms, green algae, and invertebrates' biomasses) considering the interactions between benthic components and environmental covariates (flow, conductivity, pH, water temperature, nitrate, phosphate) evolving through time, as follows:

$$X_t = BX_{t-1} + U + C_{Ct} + W_t \quad W_t \sim MVN(0, Q) \quad (1)$$

$$Y_t = ZX_t + V_t \quad V_t \sim MVN(0, R) \quad (2)$$

with $X_t$ a matrix of states at time $t$, $Y_t$ a matrix of observations at time $t$, $W_t$ a matrix of process errors (multivariate normally distributed with mean 0 and variance $Q$), $V_t$ is a matrix of observation errors (normally distributed with mean 0 and variance $R$). $Z$ is a matrix linking the observations $Y_t$ and the correspondent state $X_t$. $B$ is an interaction matrix with inter-specific interaction (diatom and green algae) and with invertebrate strengths, $C_t$ is a matrix of environmental variables (flow, conductivity, pH, water temperature, nitrate, phosphate) at time $t$. $C$ is a matrix of coefficients indicating the effect of $C_t$ to states $X_t$. $U$ describes the mean trend. We computed a total of 12 models from the most complete to the simplest,

the best-fitting model was identified as having the lowest Akaike Information Criterion adjusted for small sample sizes (AICc)[14,30]. To detect structural breaks in cyanobacteria biomass time series we calculated the differences between the smoothed state estimates at time $t$ and $t$-1 based on the multivariate models. Sudden changes in the level were detected when the standardized smoothed state residuals exceed the 95% confidence interval for a t-distribution. We estimated the strength of environmental variables on cyanobacteria biomass and fitted models independently for each stream reach.

To analyze cyanobacteria biomass across a gradient of flow alterations we compared weekly paired data ($n = 1456$) from upstream and downstream sites (i.e., at 10 m and 500 m). We thus calculated how much downstream site(s) biomass changed in comparison to upstream site biomass and assigned a factor for the increase or decrease. We determined the relative fraction of the instantaneous upstream flow in the downstream site measured within a 30-min time-step. We applied the same analysis to data from experiments obtained on the web search. We applied the Ramer–Douglas–Peucker (RDP) algorithm to find a breakpoint (ε lower distance to breakpoint) and the best line of fit for the local and global survey data distribution, we used the kmlShape-R package [48].

**Reporting summary**. Further information on research design is available in the Nature Research Reporting Summary linked to this article.

## Data availability
The datasets generated during and/or analysed during the current study are available from: https://doi.org/10.5281/zenodo.6410466.

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

## Acknowledgements

The development of this work was part of the project "Développer des solutions pour la gestion durable et adaptative des ressources en eau dans les páramos de la ville de Quito (Équateur)" - CHALPI-FLOW - funded by the Agence Française pour le Développement (AFD), in collaboration with the French Institute for a sustainable Development (IRD, convention no 2017000345). The data collection for monitoring was funded by the Water Fund of Quito FONAG and Quito's water utility EPMAPS. We would like to thank the reviewers that contributed to the improvement of our manuscript.

## Author contributions

D.R.L., M.T.W., A.S.F., B.D.B., R.O., and O.D. designed research; D.R.L., R.O., and D.G.Z. performed research and collected data; D.R.L., D.G.Z., S.C.F., and O.D. analyzed data; D.R.L. and O.D. led the writing with contributions from all co-authors.

## Competing interests

The authors declare no competing interests.
