## [Peer Review File · Communications Biology]

Reviewers' comments:

Reviewer #1 (Remarks to the Author):

This paper presents data on flow induced shifts in benthic cyanobacteria biomass. The experiment based on manipulating flow in a stream to examine how the benthic algal biomass and community structure changes as result. This is an area with few studies and from that perspective is interesting. The design is good and this makes for an interesting study for this area with local and some international interest. The paper is written well and has interesting outcomes.

One major issue with the paper is the lack of differentiation by the authors between benthic and planktonic cyanobacterial communities. Many of the references to discuss the cyanobacteria issue and also in the global literature survey were done on planktonic cyanobacterial blooms which are totally different to benthic algal growths/blooms. The factors controlling benthic algal biomass and community structure (scouring velocity, shear stress, nutrient movement around benthic mats etc.) are very different to planktonic algae (thermal stratification in pools, buoyancy regulation, Zeu:Xmix, translocation etc.). So I do not think that you can compare the two. Also the relevant references to benthic algae are very different to the planktonic literature and the risks of cyanobacteria in this paper have really been framed in the planktonic model, whereas the risks from benthic algae are quite different and need to be clearly articulated. From a drinking water perspective there is only a risk if benthic algae are dislodged or toxins are actively released into the water. Some good work in this area has been done in New Zealand at the Cawthron Institute.

I do not think that there are enough papers based on benthic algae responses to flow to be able to do an estimation of the amount of flow reduction that causes a change in the community to give a worldwide perspective. Only a few that I can see in Table S1 are based on benthic algae - the rest are planktonic algae and due to the different factors that influence blooms I do not think you can compare the two. This greatly limits the applicability of the flow removal suggestions by the authors to wider audiences. However the experiment is neat though only relevant to a particular region and benthic algae.

I suggest the authors reframe their manuscript to the published literature on benthic algae and see if there is enough information to do a similar analysis as attempted with the data in Table S1. If there is not the data available to link the local study to benthic algae / flow changes in rivers, then the study is still good, but remains local in context.

Reviewer #2 (Remarks to the Author):

Review comments, "Flow-induced shifts in river cyanobacteria biomass: A whole ecosystem experiment"

These authors report results of an experimental, ~25-week streamflow reduction and subsequent flow re-instatement in an Andean stream for the purpose of assessing the response by benthic cyanobacteria, which are expected to increase in biomass at lowered flows. The authors couple their experimental results with weekly measurements of cyanobacteria biomass above and below water withdrawals at 21 sites in 7 streams, over 2 years, and also to literature values for cyanobacterial response to streamflow reduction. The experimental results provide evidence for a 'tipping point' of ~40-60% flow reduction at which cyanobacterial biomass increases compared to baseline values, a level also supported by field measurements and literature values.

I think this is an important paper. Algal blooms are one potential outcome of prolonged and severe streamflow reduction, however actual algal responses during low-flows likely depends on nutrient levels, light availability, algal and grazer community composition and perhaps other context-specific factors. In any case, there seems to be a lot of variability reported from field observations. Controlled field experiments are critical to improving our understanding of low-flow effects on stream biota. Here, the authors implement an elegant manipulation, reducing stream flow in the experimental reach by incrementally larger amounts at a weekly time-step, and measuring

responses of benthic biota in a before-after, control-impact design. Their reverse experiment, incrementally restoring flows to assess the percent of baseline flow associated with return to low cyanobacteria biomass, provides compelling evidence for the authors' inferred influence of flow reduction.

My specific comments all pertain to clarifying details, and in some cases may simply reflect my misunderstanding.

1. Line 77 – this statement of results shows positive effects of temperature and nitrate on cyanobacteria biomass, but also a positive effect of streamflow on biomass, even though the first two variables were inversely correlated with streamflow, and reduced streamflow is the inferred trigger for increasing cyanobacteria biomass. How should one interpret this apparent positive effect of flow in the fitted model?
2. Lines 83-84 – Why not also show values for mean biomass of diatoms and green algae in Table 1? Additionally, I would appreciate some discussion of why cyanobacteria were stimulated but green algae were not. Was this expected a priori?
3. Also, what mechanism(s) caused increased nitrate when flows were diverted?

Minor editorial suggestions or questions:

4. Line 27 – I'm unsure what is meant by the sentence that includes "their relatively small instantaneous water volume"; reword?
5. Lines 50-51 – I understand the entire flow-diversion and flow reinstatement experiment to have spanned ~0.5 year; this wording implies to me that the experiment spanned a year. Clarify?
6. Line 50 – "downstream 1 reach"?
7. Line 252 – I suggest clarifying the components of the X matrix in the MARSS models. Specifically, X includes state measurements for all four benthic components (cyanobacteria, green algae, diatoms, invertebrates)?

Author response to the Referees

Title: Flow-induced shifts in river cyanobacteria biomass: A whole ecosystem experiment

Authors: Daniela Rosero-López^{1*}, M. Todd Walter¹, Alexander S. Flecker², Bert De Bièvre³, Rafael Osorio⁴, Dunia González-Zeas⁵, Sophie Cauvy-Fraunie⁶, Olivier Dangles⁵.

Comments (blue) are numbered from **1 to 16** and following each comment are responses from co-authors indicated by **R#**. In *Italic* is the text either edited or incorporated into the manuscript.

Response to Referee # 1(Phytoplank ecology): Remarks to author

1. This paper presents data on flow induced shifts in benthic cyanobacteria biomass. The experiment based on manipulating flow in a stream to examine how the benthic algal biomass and community structure changes as result. This is an area with few studies and from that perspective is interesting. The design is good and this makes for an interesting study for this area with local and some international interest. The paper is written well and has interesting outcomes.

R1. We appreciate the Reviewer finds the design of our experiment good and that it makes an interesting study with local and some international interest. Indeed, our main goal as highlighted by the reviewer is to examine how the benthic algal biomass changes in response to manipulated flow and how this affects the community structure. We value the paper was captured completely and in recognized as an important contribution.

Response to Referee #1: Specific comments

2. One major issue with the paper is the lack of differentiation by the authors between benthic and planktonic cyanobacterial communities.

R2. This is an important observation the Reviewer has kindly point out. Our main intention was to convey the message that benthic algae proliferation in rivers are as common and frequent as planktonic cyanobacterial communities are in big rivers and lakes. We initially used both communities to state the conditions that imperils freshwaters, notwithstanding, we have addressed this observation targeting only benthic algae in streams with special focus on water for human consumption.

3. Many of the references to discuss the cyanobacteria issue and also in the global literature survey were done on planktonic cyanobacterial blooms which are totally different to benthic algal growths/blooms. The factors controlling benthic algal biomass and community structure (scouring velocity, shear stress, nutrient movement around benthic mats etc.) are very different to planktonic algae (thermal stratification in pools, buoyancy regulation, Zeu:Xmix, translocation etc.). So, I do not think that you can compare the two. Also, the relevant

references to benthic algae are very different to the planktonic literature and the risks of cyanobacteria in this paper have really been framed in the planktonic model, whereas the risks from benthic algae are quite different and need to be clearly articulated.

R3. As mentioned in the above observation, we have redirected our framework to benthic algae proliferation in rivers and considered literature references and data to relate cyanobacterial increase with flow. We framed our study in cyanobacteria relation to hydrology (specifically streamflow). We conducted a thorough targeted search that allowed us to obtain data (n=25) where streamflow is considered a driver of cyanobacteria and associated toxins, increase in streams.

We have changed Figure 1, Figure 4, and Table S1 and incorporated this information in the Supplementary Material References.

In Figure 1d, we have changed the categories, originally including a “neutral” response, to two categories reporting either increase or decrease in cyanobacteria biomass according to flow reduction. We classified our findings of increase in qualitative and quantitative responses and used the latter to plot Figure 4b. Figure 1d has a new distribution of findings around the world from the updated literature search of benthic cyanobacteria in running waters also presented in supplementary references.

In *italic* is the text changed in the Figure 1d legend.

Figure 1. Cyanobacteria relations to flow in rivers. a) Experimental flow manipulations in a free-flowing high-altitude stream reach, b) using a series of v-notch weirs to reduce natural flow in fixed percentages, c) showed an increase in cyanobacteria levels, d) *worldwide studies (circles, n = 80) reporting cyanobacteria biomass decreases (green), quantitative increase (yellow), and qualitative increase (red) with respect to flow reduction.*

In Figure 4, panel b) shows only quantitative data used to calculate a cyanobacteria biomass increase factor in relation to the relative percentage of flow reduction. In *italics* is the text changed in the figure legend.

Figure 4. Cyanobacteria increase factor according to the relative percentage of flow reduction. a) Measurements (gray circles) of paired cyanobacteria-flow monitoring data ($n = 697$) at the time of the sample in one location upstream of the water intake and two locations downstream water intakes in seven streams from the water supply system; experimental results (blue circles) of cyanobacteria increase with targeted flow reductions including variations \pm SE from temporal replicates within flow reduction (black lines). Rammer-Douglas-Peucker model (RDP) (green line) fitted to monitoring data showing a breakpoint for cyanobacteria increase with a 40% flow reduction. **b)** A global survey of cyanobacteria-flow data ($n = 25$) showing a distribution fitted with RDP model: cyanobacteria increase after a breakpoint of 50% flow reduction (blue line).

4. From a drinking water perspective there is only a risk if benthic algae are dislodged, or toxins are actively released into the water. Some good work in this area has been done in New Zealand at the Cawthron Institute.

R4. We considered this observation highly valuable to incorporate to achieve a deeper understanding of the threaten that represents benthic cyanobacteria proliferation in running waters. Safe drinking water has been recognized and called out by the scientific community as a major issue to address in the face of climate change and anthropogenic activities for the next 10 years. We incorporated the cutting-edge analysis done in the Cawthron Institute by Wood et al. 2006, 2018, and 2020, and we used valuable data to show the importance of leaving water flowing in streams.

5. I do not think that there are enough papers based on benthic algae responses to flow to be able to do an estimation of the amount of flow reduction that causes a change in the community to give a worldwide perspective. Only a few that I can see in Table S1 are based on benthic algae - the rest are planktonic algae and due to the different factors, that influence blooms I do not think you can compare the two. This greatly limits the applicability of the flow removal suggestions by the authors to wider audiences. However, the experiment is neat though only relevant to a particular region and benthic algae.

R5. As mentioned above, we have targeted the literature survey to identify benthic algae and cyanobacteria proliferation in rivers and streams. Most of the newly incorporated literature ($n = 80$) are findings related in one way or another to drinking water in the first place, and then aesthetics and recreation. We have eliminated planktonic conditions in lakes and replaced with benthic cyanobacteria cases reporting changes with flow. Table S1 details conditions and data used to calculate the factor of increase in cyanobacteria biomass according to the reduction of flow from a previous condition. Also, we have incorporated a vast number of

references that report qualitative data around the world and presented in the Supplementary References.

Table S1. Research used ($n = 25$) to calculate the percent of flow remaining in streams and a factor of cyanobacteria increase or decrease according to reported baseline conditions (either temporal or spatial).

Freshwater system	Landscape	Elevation (m) a.s.l.	Location / Country	Discharge (m ³ /s)	Cyanobacteria dominant taxa	Response to flow reduction	Reference
Darling	Floodplain	110	Mungindi	9.72	Anabaena	Increase	Bowling and Baker 1996
Barwon	Floodplain	295	Wentworth	10.76	Anabaena	Increase	Bowling and Baker 1996
Darling	Floodplain	35	Bourke	5.2	Anabaena	Increase	Mitrovic et al. 2003
Darling	Floodplain	110	Walgett	1.2	Anabaena	Increase	Mitrovic et al. 2003
Darling	Floodplain	75	Wilcannia	2.3	Anabaena	Increase	Mitrovic et al. 2003
Darling	Floodplain	119	Brewarrina	5.1	Anabaena	Increase	Mitrovic et al. 2003
Barwon-Darling	Floodplain	78	Barwon	2.5	Anabaena	Increase	Mitrovic et al. 2005
Okuku	Grassland	654	New Zealand	4.64	Phormidium	Increase	Suren et al. 2003
Waipara	Grassland	105	New Zealand	3.06	Phormidium	Increase	Suren et al. 2003
Murrumbidgee	Floodplain	55	Maud	850	Anabaena	Increase	Webster et al. 2000
Murray-Darling	Floodplain	3.6	Blanchetown	11.57	Lyngbya	Increase	Burns and Walker 2000
Colorado river	Great basin desert	944	Lees Ferry	414	Oscillatoria	Increase	Benenati et al. 2000
Llobregat	Mediterranean	N/A	Llobregat	0.25	Oscillatoria	Increase	Sabater et al. 2003
Llobregat	Mediterranean	N/A	Llobregat	0.17	Oscillatoria	Increase	Vilalta et al. 2003
Guadiana	Estuarine	20	Pulo de Lobo	220	Myrosystis	Decrease	Domingues et al. 2005
Wainuomata	Floodplain	0	Manuka	0.95	Phormidium	Increase	Heath et al. 2011
Mangaroa	Floodplain	220	Mangaroa	3.18	Phormidium	Increase	Heath et al. 2011
Hutt	Floodplain	0	Boulcott	20.4	Phormidium	Increase	Heath et al. 2011
Hutt	Floodplain	0	Silverstream	26.14	Phormidium	Increase	Heath et al. 2011
Hutt	Floodplain	122	Whakatikei	14.15	Phormidium	Increase	Heath et al. 2011
Hutt	Floodplain	145	Akatarwa	10.22	Phormidium	Increase	Heath et al. 2011
Hutt	Floodplain	140	Te Maura	11.14	Phormidium	Increase	Heath et al. 2011
Hutt	Floodplain	232	Hutt	15.21	Phormidium	Increase	Heath et al. 2015
Murray	Floodplain	N/A	Morgan	767	Anabaena	Increase	Maier et al. 2004
Mississippi	Upland waters	244	USA	655	Dolichospermum	Increase	Giblin and Gerrish 2020

6. I suggest the authors reframe their manuscript to the published literature on benthic algae and see if there is enough information to do a similar analysis as attempted with the data in Table S1. If there is not the data available to link the local study to benthic algae / flow changes in rivers, then the study is still good, but remains local in context.

R6. We value this observation highly since it encouraged us to frame our findings in an area that calls special attention for environmental flow management. We replaced the original Table S1 and incorporated several studies from recent reviews on benthic cyanobacteria proliferation in running waters (Espinosa et al. 2020, Wood et al. 2020). We carefully selected observations from big rivers reporting benthic algae despite the presence of phytoplankton bloom observations.

Response to Referee #2 (Environment Science and Ecology): Remarks to author

7. These authors report results of an experimental, ~25-week streamflow reduction and subsequent flow re-instatement in an Andean stream for the purpose of assessing the response by benthic cyanobacteria, which are expected to increase in biomass at lowered flows. The authors couple their experimental results with weekly measurements of cyanobacteria biomass above and below water withdrawals at 21 sites in 7 streams, over 2 years, and also to literature values for cyanobacterial response to streamflow reduction. The experimental results provide evidence for a ‘tipping point’ of ~40-60% flow reduction at which cyanobacterial biomass increases compared to baseline values, a level also supported by field measurements and literature values.

R7. We appreciate the reviewer clear description of our findings from the experiment at the ecosystem scale to the comparison of nearby observations and global reporting on cyanobacteria presence in running waters.

8. I think this is an important paper. Algal blooms are one potential outcome of prolonged and severe streamflow reduction, however actual algal responses during low-flows likely depends on nutrient levels, light availability, algal and grazer community composition and perhaps other context-specific factors. In any case, there seems to be a lot of variability reported from field observations.

R8. This observation points-out the cumbersome of responses reported in highly dynamic systems such as tropical Andean streams. Our intention was to identify the extent of this variability that could be attributable to streamflow. Our detailed manipulation of flow considering the natural variability in nearby streams as well as in the stream subjected to the manipulation, allowing us to identify changes that although occurring naturally (nutrient levels, light availability, and grazer community composition) have shown a response to the changes induced by flow manipulation.

9. Controlled field experiments are critical to improving our understanding of low-flow effects on stream biota. Here, the authors implement an elegant manipulation, reducing stream flow in the experimental reach by incrementally larger amounts at a weekly time-step, and measuring responses of benthic biota in a before-after, control-impact design. Their reverse experiment, incrementally restoring flows to assess the percent of baseline flow associated with return to low cyanobacteria biomass, provides compelling evidence for the authors' inferred influence of flow reduction.

R9. We appreciate the Reviewer recognition of the effort that represented placing an experiment design of this detail, particularly the difficult to manipulate streamflows in a highly hydrological dynamic system. Our design was possible to set up after several laboratory and field testing as well as observations of water abstraction operation in high altitude streams. Our manipulation structure provided us the opportunity to control streamflows and at the same time measure ecosystem responses that we mildly understand in these streams.

Response to Referee # 2: Specific comments

10. Line 77 – this statement of results shows positive effects of temperature and nitrate on cyanobacteria biomass, but also a positive effect of streamflow on biomass, even though the first two variables were inversely correlated with streamflow, and reduced streamflow is the inferred trigger for increasing cyanobacteria biomass. How should one interpret this apparent positive effect of flow in the fitted model?

R10. The fitted model assessed the condition of cyanobacteria biomass increase, therefore the direction of the effect of flow on cyanobacteria was considered positive and we used the same criteria for the increase in temperature and nitrate concentration. As the reviewer observed the positive effect of flow reduction on nitrate concentration increase opens the question about the observed lag response. We have modified the following paragraph in the results and incorporated a remark in the discussion:

Line 81: *A multivariate, autoregressive state-space analysis revealed a significant positive effect of flow on cyanobacteria biomass related with the effect on temperature and nitrate*

concentration increase ($C_{Flow \rightarrow Cyano} = 0.031$, $C_{Temp \rightarrow Cyano} = 0.016$, $C_{NO3^- \rightarrow Cyano} = 0.023$, $AIC_c = 161.1$; see *Materials and Methods*).

Line 149: *It is likely that more prolonged flow reductions would prevent the system from fully recovering between consecutive flow disturbances, with potentially severe consequences on water quality indicated by the nitrate concentration increase (8, 21, 47).*

11. Lines 83-84 – Why not also show values for mean biomass of diatoms and green algae in Table 1? Additionally, I would appreciate some discussion of why cyanobacteria were stimulated but green algae were not. Was this expected a priori?

R11. We appreciate this observation, and it has been incorporated in Table 1.

Table 1. Mean abiotic and biotic variables during the experimental phases (BL: baseline, FR: flow reduction, FI: flow re-instate as flow recovery) in four experimental reaches in two streams (manipulated and control). * Indicates significant differences in parameter value when compared to values from the baseline conditions before alteration (significant *p* values at 0.05 alpha level from a paired one-tail t-test).

	Manipulated stream						Control stream					
	Upstream site			Downstream site			Upstream site			Downstream site		
	BL	FR	FI	BL	FR	FI	BL	FR	FI	BL	FR	FI
Discharge ($m^3 \cdot s^{-1}$)	0.188	0.211	0.159	0.188	0.087*	0.103*	0.019	0.021	0.023	0.019	0.022	0.024
(% CV)	57	82	69	57	99	91	19	30	24	18	29	22
Temperature ($^{\circ}C$)	7.4	7.7	8.1	7.4	10.1*	9.5*	6.9	6.9	7.1	7.1	6.9	7.1
(min - max)	4.1 - 10.5	4.7 - 10.6	5.1 - 11.1	4.3 - 10.5	5.6 - 11.7	5.1 - 11.1	4.1 - 9.4	4.1 - 9.3	4.2 - 9.6	4.2 - 9.5	4.1 - 9.5	4.1 - 9.3
pH	6.9	6.8	6.8	6.9	7.2	6.8	6.8	6.8	7.0	6.8	6.9	6.9
(\pm SD)	0.3	0.3	0.4	0.5	0.6	0.4	0.3	0.4	0.3	0.4	0.3	0.2
Conductivity ($\mu S \cdot cm^{-1}$)	40.03	44.05	42.01	44.77	48.23	43.07	38.55	38.39	41.01	39.74	37.88	40.71
(\pm SD)	2.33	3.21	2.81	2.88	4.91	3.71	1.55	2.11	1.65	1.56	2.11	1.65
Dissolved Oxygen (mg. L)	7.91	7.75	8.24	7.92	7.69	7.88	8.12	7.89	7.73	7.79	7.99	7.54
(min - max)	6.1 - 9.9	6.9 - 9.7	6.2 - 9.9	6.6 - 10.3	6.2 - 9.2	6.1 - 9.4	6.3 - 9.4	6.5 - 9.9	6.4 - 10.1	6.3 - 9.8	6.5 - 9.9	6.5 - 10.1
Nitrate (mg. L ⁻¹)	0.34	0.24	0.34	0.26	0.36*	0.48*	0.16	0.17	0.18	0.16	0.16	0.18
(\pm SD)	0.04	0.07	0.02	0.06	0.13	0.05	0.05	0.03	0.09	0.06	0.03	0.09
Phosphate ($\mu g \cdot L^{-1}$)	0.07	0.07	0.09	0.09	0.08	0.08	0.05	0.06	0.05	0.07	0.07	0.08
(\pm SD)	0.004	0.005	0.003	0.005	0.005	0.006	0.002	0.002	0.004	0.005	0.003	0.006
Cyanobacteria (μg Chl-a. cm^{-2})	1.86	1.89	1.63	1.22	3.95*	3.23*	0.99	1.04	1.08	0.95	0.97	1.05
(\pm SD)	0.39	0.39	0.32	0.45	2.12	2.25	0.07	0.21	0.22	0.09	0.19	0.17
Diatoms (μg Chl-a. cm^{-2})	2.12	2.04	1.97	1.92	2.67*	2.86*	1.48	1.58	1.45	1.32	1.62	1.52
(\pm SD)	0.36	0.43	0.38	0.53	1.04	1.09	0.29	0.34	0.34	0.14	0.32	0.31
Green algae (μg Chl-a. cm^{-2})	2.08	1.43	1.64	1.95	1.18	0.94	1.26	1.07	1.01	1.24	0.95	0.91
(\pm SD)	0.33	0.39	0.40	0.43	0.48	0.54	0.13	0.44	0.34	0.15	0.18	0.21
Invertebrates (g DM. m^{-2})	3.85	4.01	4.61	3.94	3.63	4.11	2.99	2.34	2.94	2.83	3.02	2.54
(\pm SD)	0.51	0.69	0.97	0.79	1.02	0.89	0.54	0.43	0.78	0.48	0.64	0.73

The relevance for accounting cyanobacteria proliferation in our research was to identify flow standards for water abstractions and reduce suitable conditions for harmful algae bloom in drying downstream reaches. Our ~1.5-year monitoring revealed a prevalent dominance of cyanobacteria in drying downstream reaches that ultimately lead to a supply system in construction. Also, the water supply company concern for cyanobacteria was based on a reported presence of *Dolichospermum* in the system, a sensitive matter for their operation.

We have incorporated the following lines in the paragraph:

Line 129: *Cyanobacteria is of particular interest as its increase/dominance in the benthic community jeopardizes the quality of potable water. A 40 to 60% flow reduction coincides with thresholds found for other organisms (e.g., invertebrates) (44) and is well above the legal environmental flow thresholds recommended in many countries (e.g., 10% in Ecuador, 25% in Colombia, and 10% in Chile).*

12. Also, what mechanism(s) caused increased nitrate when flows were diverted?

R12. This observation is of particular interest to us as it reflects on the oligotrophic conditions of these streams. The lag in the response of nitrate concentration under flow reduction is yet to investigate and we look forward to unveiling other mechanisms associated with microbes and temperature. We have this paragraph:

Line 152: *A longer time flow reduction will elucidate the underlying mechanism for nitrate increase as diatoms and green algae recovers. Such pressures may also have environmental consequences beyond the threshold's timeline, with underestimated effects on the system (48).*

13. Line 27 – I'm unsure what is meant by the sentence that includes “their relatively small instantaneous water volume”; reword?

R13. We have modified the paragraph as follows:

Line 24: *The understanding of ecosystem state shifts in running waters could provide early-warning signals for managing water resources at watershed scales that, in mountain systems, often include lakes and reservoirs. Indeed, streams have relatively small instantaneous water volume when compared to lakes and may therefore respond more quickly to environmental changes (even though the continuous water renewal may also increase stream's resistance to stress).*

14. Lines 50-51 – I understand the entire flow-diversion and flow reinstatement experiment to have spanned ~0.5 year; this wording implies to me that the experiment spanned a year. Clarify?

R14. Yes, the observation from the reviewer is correct. We have modified the text as follows:

Line 50: *The stream analysis consisted of three phases: (1) establishment of baseline conditions (BL) under unaltered flow (~ 1.5 years); (2) experimental diversion of flow, inducing systematic flow reductions in the downstream reach (~ 0.3 years), and (3) gradual reset to initial flow (FI) conditions in the downstream reach (~ 0.3 years) (SI Appendix, Fig. S1).*

15. Line 50 – “downstream 1 reach”?

R15. Changed

16. Line 252 – I suggest clarifying the components of the X matrix in the MARSS models. Specifically, X includes state measurements for all four benthic components (cyanobacteria, green algae, diatoms, invertebrates)?

R16. We have incorporated the suggestion from the reviewer to make clearer the components of X matrix:

Line 259: *The state processes X_t includes state measurements for all four benthic components (cyanobacteria, diatoms, green algae, and invertebrates' biomasses) considering the interactions between benthic components and environmental covariates (flow, conductivity, pH, water temperature, nitrate, phosphate) evolving through time, as follows.*

Reviewers' comments:

Reviewer #1 (Remarks to the Author):

The paper has improved but still has one issue that needs to be fixed in that the authors have not accurately understood which papers refer to planktonic cyanobacteria and which refer to benthic cyanobacteria. This needs to be addressed before the paper is suitable for publication. Below I give the papers that are not benthic cyanobacteria, but are planktonic cyanobacteria. These should be removed from the literature review. This may leave the authors with not enough papers to make this analysis worthwhile, or perhaps they may just use this to illustrate the work done with benthic algae. Overall in the paper the authors should continue to make it clear they are referring to benthic cyanobacteria to avoid confusion with the very different planktonic blooms.

Other comments

Title - change to "... shifts in benthic river cyanobacteria ..."

Abstract - line 2, To determine benthic cyanobacteria ...

Abstract line 5. Benthic cyanobacteria greatly increased

Abstract - Line 8 flow-benthic algal measurements....

Abstract line 10 - global literature review will need to be redone for only benthic algal studies - see below.

Line 14. remove "fine"

26 - reservoirs that could be manipulated.

29. Be good to add a reference to support this point.

39 limited in benthic cyanobacteria of running waters...

42. shifts in benthic cyanoHABs and

63. quantitative benthic cyanobacteria biomass

Fig 1 - I believe the authors are still capturing planktonic cyanobacteria as well as benthic. Again see my comments below about which papers in the table are planktonic and which are benthic.

96. significant benthic cyanobacteria biomass...

100 - will need to re-run the literature review based survey based on only benthic algae.

Figure 4. Benthic cyanobacteria increase factor...global survey of benthic cyanobacteria flow data

134. Not sure of this line...as cyanobacteria proliferation in streams and rivers can threaten downstream lakes and

135 reservoirs (23, 45, 46), - how do benthic algae effect downstream lakes? By being scoured and moving down to them? Need to explain this better and have references to support the issue.

136 - benthic cyanobacteria biomass

140 triggers benthic alga proliferation...

210 "benthic" or "benthic cyanobacteria " should be a term for the review to remove planktonic papers.

Table S1 has many studies that are planktonic and not benthic. These include the below, but may also include some of the others. These should be removed from the analysis.

Bowling and Baker 1996

Mitrovic et al. 2003

Mitrovic et al. 2005

Webster et al. 2000

Domingues et al. 2005

Maier et al. 2004

Giblin and Gerrish 2020

Any studies with phytoplankton cannot be used as the controls on their biomass are totally different to benthic algae.

Due to this the lit revue survey will need to be re-run. I am also concerned that there may not now be enough papers to support this analysis. This may mean that this analysis is removed, and the remaining papers could be refereed to in the discussion and to highlight the lack of work on benthic algae in streams and rivers.

Reviewer #2 (Remarks to the Author):

The authors have responded to all of my review comments (on the previous ms version) and I do not have any additional suggestions other than a few minor edits (listed below). I think the paper reports an elegant field experiment that, along with associated data, presents a compelling case for effects of streamflow diversion on cyanobacteria biomass accumulation. I think this is an important contribution to environmental flow science.

Suggestions for minor edits:

Line 14 – suggest replacing “fine” with “detailed”

Line 19 – suggest inserting “a” - “may also increase a stream’s resistance to stress”

Line 37 – consider replacing “proliferated” with “widespread”

Line 52 – suggest not deleting “(FR)” as marked, unless you also delete “(FI)” in line 53

Line 58 – suggest replacing “Moreover” with “Additionally”

Figure 4 caption – change the estimated breakpoint from “50%” to “58%”? (last sentence)

Line 121 – suggest “These temperature and nutrient increases with flow reduction confirm that...”

Lines 151-152 – Not sure I understand the added sentence that begins “A longer time flow reduction will elucidate...”. “might elucidate”? I’m unclear what is meant by “as diatoms and green algae recover”; my understanding is that green algae did not respond to the flow manipulation.

Line 173 – replace “previously to conduct” with “prior to conducting”

Line 182 – correct spelling “evaluate”

Lines 260, 263 – replace “invertebrate’s” with “invertebrate”

Author response to the Referees

Title: Flow-induced shifts in benthic river cyanobacteria biomass: A whole ecosystem experiment

Authors: Daniela Rosero-López^{1*}, M. Todd Walter¹, Alexander S. Flecker², Bert De Bièvre³, Rafael Osorio⁴, Dunia González-Zeas⁵, Sophie Cauvy-Fraunié⁶, Olivier Dangles⁵.

Comments (blue) are numbered from **1 to 33** and following each comment are responses from co-authors indicated by **R#**. In *Italic* is the text either edited or incorporated into the manuscript.

Response to Referee # 1: Remarks to author

1. The paper has improved but still has one issue that needs to be fixed in that the authors have not accurately understood which papers refer to planktonic cyanobacteria and which refer to benthic cyanobacteria. This needs to be addressed before the paper is suitable for publication. Below I give the papers that are not benthic cyanobacteria but are planktonic cyanobacteria. These should be removed from the literature review. This may leave the authors with not enough papers to make this analysis worthwhile, or perhaps they may just use this to illustrate the work done with benthic algae. Overall, in the paper the authors should continue to make it clear they are referring to benthic cyanobacteria to avoid confusion with the very different planktonic blooms.

R1. This is a good point. In the previous version, we included planktonic and benthic cyanobacteria as hydrological drivers (e.g., streamflow) can be involved in the proliferation of both community types. But we agree that the underlying processes and impacts are different and that it is better to focus on benthic cyanobacteria to avoid confusion. In the revised version we have removed all references to planktonic cyanobacteria in running waters and found new research studies that reported toxins from benthic cyanobacteria associated with flow reduction (see the final list in Table S1). Our new data set on benthic cyanobacteria now includes 53 study sites (instead of 80). We have re-run the analysis that still appears sufficiently robust to provide sound conclusions (see new Fig. 4b).

Responses to Referee #1: Specific comments

2. Title - change to "... shifts in benthic river cyanobacteria ...".

R2. Changed:

*Title: Flow-induced shifts in **benthic** river cyanobacteria biomass: A whole ecosystem experiment*

3. Abstract - line 2, To determine benthic cyanobacteria ...

R3. Changed:

*To determine **benthic** cyanobacteria regime shifts in a potable water supply system in the tropical Andes*

4. Abstract line 5. Benthic cyanobacteria greatly increased

R4. Changed:

Benthic cyanobacteria greatly increased with a 60% flow reduction and this tipping point was related to water temperature and nitrate concentration increases

5. Abstract - Line 8 flow-benthic algal measurements....

R5. Changed:

We supplemented our experiment with a regional survey collecting > 1450 flow-benthic algal measurements at streams varying in water abstraction levels.

6. Abstract line 10 - global literature review will need to be redone for only benthic algal studies - see below.

R6. We have revised and updated the global literature review selecting **benthic** cyanobacteria responses to flow.

7. Line 14. remove "fine"

R7. Edited.

*Our **detailed** comprehension of ecosystem state shifts in lakes has been possible through ecosystem-scale experiments (1, 2, 7, 10, 11).*

8. Line 26. reservoirs that could be manipulated.

R8. Incorporated:

Line 24: *The understanding of ecosystem state shifts in running waters could provide early-warning signals for managing water resources at watershed scales that, in mountain systems, often include lakes and reservoirs **that could be manipulated**.*

9. Line 29. Be good to add a reference to support this point.

R9. We have included reference 22 to support this statement:

Line 27: *Indeed, streams have relatively small instantaneous water volume when compared to lakes and may therefore respond more quickly to environmental changes (22) (even though the continuous water renewal may also increase a stream's resistance to stress).*

(22) C. Quiblier, S. Wood, I. Echenique-Subiabre, M. Heath, A. Villeneuve, J.F. Humbert. A review of current knowledge on toxic benthic freshwater cyanobacteria–ecology, toxin production and risk management. *Water research* 47(15): 5464-5479. (2013).

10. Line 39 limited in benthic cyanobacteria of running waters

R10. Changed:

Line 39: *While planktonic CyanoHABs have been well addressed in lakes and large impounded rivers (e.g., 3, 7, 11), our knowledge is much more limited in **benthic** cyanobacteria of running waters, where ecological drivers (e.g., temperature, pH, dissolved oxygen) of **benthic** cyanoHABs and dynamics of system transitions are poorly understood (see 22, 27, 28).*

11. Line 42. shifts in benthic cyanoHABs

R11. Changed Line 41 (see R10).

12. Line 63. quantitative benthic cyanobacteria biomass

R12. Changed:

Lines 61 – 64: *These dual approaches enabled us to evaluate whether experimental results were consistent with the **benthic** cyanobacteria response to permanent flow alteration across a gradient of flow reduction in the same study area (from 98% to 23%). Also, we performed a literature review of qualitative and quantitative benthic cyanobacteria biomass response to changes in flow levels (see Fig. 1d).*

13. Fig 1 - I believe the authors are still capturing planktonic cyanobacteria as well as benthic. Again, see my comments below about which papers in the table are planktonic and which are benthic.

R13. The reviewer is correct as we focused the research on running waters that presented planktonic cyanobacteria blooms too. We have eliminated these references and the study sites we initially considered. The updated Figure 1d presents benthic cyanobacteria relations with flow reduction and now includes 53 study sites (instead of 80 in the previous version) from which three reported decreases of benthic cyanobacteria with flow reduction. Among them, 20 studies presented qualitative data, and 33 studies presented quantitative data, which were used to produce Figure 4b.

Figure 1. Benthic cyanobacteria relations to flow in rivers. a) Experimental flow manipulations in a free-flowing high-altitude stream reach, **b)** using a series of v-notch weirs to reduce natural flow in fixed percentages, **c)** showed an increase in benthic cyanobacteria levels, **d)** worldwide studies (circles, $n = 53$)

reporting qualitative observations of benthic cyanobacteria biomass decreases (yellow) and qualitative and quantitative observations of biomass increases (teal-blue) with respect to flow reduction. Right-downside frame show data reported for New Zealand.

14. Line 96. significant benthic cyanobacteria biomass...

R14. Edited:

Line 96: *We found no significant shifts in benthic cyanobacteria, algal and invertebrate biomass on the upstream and downstream reaches of the control stream (SI Appendix, Fig. S2a, Fig. S3b).*

15. Line 100 - will need to re-run the literature review-based survey based on only benthic algae. Figure 4. Benthic cyanobacteria increase factor...global survey of benthic cyanobacteria flow data.

R15. We have re-run the literature review based only on benthic cyanobacteria; we were able to incorporate study sites with quantitative information that allowed us to calculate the cyanobacteria biomass increase factor. The updated Figure 4b now includes 33 data points:

Figure 4. Benthic cyanobacteria increase factor according to the relative percentage of flow reduction. a) Measurements (gray circles) of paired cyanobacteria-flow monitoring data ($n = 697$) at the time of the sample in one location upstream of the water intake and two locations downstream water intakes in seven streams from the water supply system; experimental results (blue circles) of cyanobacteria increase with targeted flow reductions including variations $\pm SE$ from temporal replicates within flow reduction (black lines). Rammer-Douglas-Peucker model (RDP) (green line) fitted to monitoring data showing a breakpoint for cyanobacteria increase with a 40% flow reduction. **b)** A global survey of benthic cyanobacteria-flow data ($n = 33$) showing a distribution fitted with RDP model: cyanobacteria increase after a breakpoint of 58% flow reduction (teal blue line).

16. Line 134. Not sure of this line...as cyanobacteria proliferation in streams and rivers can threaten downstream lakes and...

R16. We have rephrased our idea to explain the potential of benthic cyanobacteria entering water systems, we also incorporated references 36 and 47:

Line 35: *Our findings, therefore, call for a revision of these recommendations to ensure sustainable water management at the watershed scale, as **benthic cyanobacteria can be scoured and moved down streams and rivers, thereby threatening lakes and reservoirs** (23, 46, 47), interconnected in potable water systems (29, 31). Stagnant waters in rivers shorelines have also been documented as a suitable habitat for floating toxin-cyanobacteria (i.e., *Anabaena* spp.), increasing the potential of entering water systems (36).*

(36) J.A. Baker, B. A. Neilan, B. Entsch, and D.B. Mckay. Identification of cyanobacteria and their toxigenicity in environmental samples by rapid molecular analysis. *Environmental toxicology*, 16(6), 472-482. (2001).

(47) S.A. Wood, J. Atalah, A. Wagenhoff, L. Brown, K. Doehring, R.G. Young, I. Hawes. Effect of river flow, temperature, and water chemistry on proliferations of the benthic anatoxin-producing cyanobacterium *Phormidium*. *Freshwater Science* 36 (1): 63-76. (2017).

17. Line 135 reservoirs (23, 45, 46), - how do benthic algae effect downstream lakes? By being scoured and moving down to them? Need to explain this better and have references to support the issue.

R17. Edited line 137, see comment **R16**.

18. Line 136 - benthic cyanobacteria biomass

R18. Changed.

19. Line 140 triggers benthic alga proliferation...

R19. Changed.

20. Line 210 "benthic" or "benthic cyanobacteria " should be a term for the review to remove planktonic papers.

R20. Changed:

Line 214: *We used ISI Web of Science, Google Scholar, and Google Search for the entries: “**benthic cyanobacteria**” + “stream” and “river”, “**benthic algal bloom**” + “flow” and available combinations (SI Appendix, Table S1). We selected papers containing information on benthic cyanobacteria and algae biomass and flow or water level measurements; specifically, we explored detailed information regarding experiments, spatial studies with upstream and downstream sites, and temporal replicates, as well seasonal associated benthic cyanobacteria blooms.*

21. Table S1 has many studies that are planktonic and not benthic. These include the below but may also include some of the others. These should be removed from the analysis. Bowling and Baker 1996; Mitrovic et al. 2003; Mitrovic et al. 2005; Webster et al. 2000 Domingues et al. 2005; Maier et al. 2004; Giblin and Gerrish 2020. Any studies with phytoplankton cannot be used as the controls on their biomass are totally different to benthic algae. Due to this the lit revue survey will need to be re-run. I am also concerned that there may not now be enough papers to support this analysis. This may mean that this analysis is

removed, and the remaining papers could be referred to in the discussion and to highlight the lack of work on benthic algae in streams and rivers.

R21. We have updated Table S1 and eliminated references and study sites in rivers that were not reporting exclusively benthic cyanobacteria. Our analysis now includes more study sites from where we were able to obtain quantitative data analyzed in Figure 4b (see comment R15). The updated Table S1 shows 53 study sites included in 20 references with the dominant benthic cyanobacteria that responded to flow reduction.

Table S1. Study sites where benthic cyanobacteria responded to flow reduction ($n = 53$), quantitative data used ($n = 33$) to calculate the percent of flow remaining in streams and a factor of benthic cyanobacteria increase or decrease according to reported baseline flow conditions (either temporal or spatial).

Freshwater system	Landscape	Elevation (m) a.s.l.	Location / Country	Discharge (m^3/s)	Cyanobacteria dominant taxa	Response to flow reduction	Reference
Murray-Darling	Floodplain	3.6	Australia	11.57	Lyngbya	Increase	Burns and Walker 2000
Colorado river	Great basin desert	944	United States	414	Oscillatoria	Increase	Benenati et al. 2000
Mataura *	Floodplain	12	New Zealand	60	Phormidium	Increase	Hamill 2001
Waikanae*	Floodplain	9	New Zealand	28	Oscillatoria	Increase	Hamill 2001
Muga*	Coastal mountains	310	Spain	109.7	Phormidium	Decrease	Aboal et al. 2002
Parrizal*	Coastal mountains	650	Spain	107.5	Phormidium	Decrease	Aboal et al. 2002
Okuku	Grassland	654	New Zealand	4.64	Phormidium	Increase	Suren et al. 2003
Waipara	Grassland	105	New Zealand	3.06	Phormidium	Increase	Suren et al. 2003
Llobregat	Mediterranean	N/A	Spain	0.25	Oscillatoria	Increase	Sabater et al. 2003
Llobregat	Mediterranean	N/A	Spain	0.17	Oscillatoria	Increase	Vilalta et al. 2004
Alharabe	Forested highlands	1000	Spain	42.5	Mycrosystis	Increase	Aboal et al. 2005
Alharabe*	Forested highlands	1000	Spain	42.5	Rivularia	Decrease	Aboal et al. 2005
Motueka*	Coastal mountains	182	New Zealand	23.3	Oscillatoria	Increase	Wood et al. 2006
Waikanae*	Coastal mountains	101	New Zealand	19.4	Oscillatoria	Increase	Wood et al. 2006
Motupiko*	Coastal mountains	100	New Zealand	11.4	Oscillatoria	Increase	Wood et al. 2006
Tikotu*	Coastal mountains	78	New Zealand	21.5	Oscillatoria	Increase	Wood et al. 2006
Waikato*	Coastal mountains	90	New Zealand	15.7	Phormidium	Increase	Wood et al. 2006
Utakura*	Coastal mountains	56	New Zealand	9.44	Mycrosystis	Increase	Wood et al. 2006
Yabba*	Floodplain	65.4	Australia	N/A	Lyngbya	Increase	Seifert et al. 2007
Brisbane*	Floodplain	213	Australia	N/A	Lyngbya	Increase	Seifert et al. 2007
Providencia*	Headwaters	1800	United States	1.82	Anabaena	Increase	Brown et al. 2008
Duff*	Headwaters	1500	United States	0.88	Anabaena	Increase	Brown et al. 2008
Bull*	Headwaters	2160	United States	8.58	Anabaena	Increase	Brown et al. 2008
Teakettle*	Headwaters	2005	United States	34.41	Anabaena	Increase	Brown et al. 2008
Fuerosos*	Mediterranean	700	Spain	0.025	Phormidium	Increase	Tornés & Sabater 2010
Wainuiomata	Floodplain	0	New Zealand	0.95	Phormidium	Increase	Heath et al. 2011
Mangaroa	Floodplain	220	New Zealand	3.18	Phormidium	Increase	Heath et al. 2011
Hutt in Boulcott	Floodplain	0	New Zealand	20.4	Phormidium	Increase	Heath et al. 2011
Hutt in Silverstream	Floodplain	0	New Zealand	26.14	Phormidium	Increase	Heath et al. 2011
Hutt in Whakatikei	Floodplain	122	New Zealand	14.15	Phormidium	Increase	Heath et al. 2011
Hutt in Akatarua	Floodplain	145	New Zealand	10.22	Phormidium	Increase	Heath et al. 2011
Hutt in Te Maura	Floodplain	140	New Zealand	11.14	Phormidium	Increase	Heath et al. 2011
Hutt	Floodplain	232	New Zealand	15.21	Phormidium	Increase	Heath et al. 2013
Li	Headwaters	740	Norway	3.52	Phormidium	Increase	Schneider et al. 2015
Eel	Coastal mountains	0	United States	260	Anabaena	Increase	Bouma-Gregson et al. 2017
Makakahi	Pastoral land use	22	New Zealand	3.18	Phormidium	Increase	Wood et al. 2017
Manawatu	Pastoral land use	45	New Zealand	73.4	Phormidium	Increase	Wood et al. 2017
Mangatainoka	Native vegetation	24	New Zealand	2.13	Phormidium	Increase	Wood et al. 2017
Oroua	Pastoral land use	32	New Zealand	7.1	Phormidium	Increase	Wood et al. 2017
Orouakeretaki	Pastoral land use	79	New Zealand	1.42	Phormidium	Increase	Wood et al. 2017
Tiraumea	Pastoral land use	80	New Zealand	7.21	Phormidium	Increase	Wood et al. 2017
Tokomaru	Native vegetation	100	New Zealand	1.25	Phormidium	Increase	Wood et al. 2017
Nederbach*	Headwaters	980	Austria	N/A	Chamaesiphon	Increase	Aigner et al. 2018
Isar*	Headwaters	980	Austria	2.2	Synschoococcales	Increase	Aigner et al. 2018
Waipara	Intensive agriculture	194	New Zealand	0.9	Phormidium	Increase	McAllister et al. 2018
Ashley	Intensive agriculture	1802	New Zealand	10.2	Phormidium	Increase	McAllister et al. 2018
Selwyn	Intensive agriculture	256	New Zealand	2	Phormidium	Increase	McAllister et al. 2018
Orari	Intensive agriculture	171	New Zealand	6.2	Phormidium	Increase	McAllister et al. 2018
Temuka	Intensive agriculture	23	New Zealand	3.4	Phormidium	Increase	McAllister et al. 2018
Opihi	Intensive agriculture	185	New Zealand	8.7	Phormidium	Increase	McAllister et al. 2018
Te ana a wai	Intensive agriculture	112	New Zealand	1.8	Phormidium	Increase	McAllister et al. 2018
Pareora	Intensive agriculture	138	New Zealand	1.4	Phormidium	Increase	McAllister et al. 2018
Ter	Mid-section	1100	Spain	0.01	Oscillatoria	Increase	Espinosa et al. 2020

* Qualitative observations

Response to Referee #2: Remarks to author

22. The authors have responded to all of my review comments (on the previous ms version) and I do not have any additional suggestions other than a few minor edits (listed below). I think the paper reports an elegant field experiment that, along with associated data, presents a compelling case for effects of streamflow diversion on cyanobacteria biomass accumulation. I think this is an important contribution to environmental flow science.

R22. We appreciate the referee recognition of the clarity of our field experiment. We have addressed the reviewer 11 specific comments and we present the changed and/or incorporated text. We hope this version is suitable for publication.

Response to Referee # 2: Specific comments

23. Line 14 – suggest replacing “fine” with “detailed”

R23. Changed.

*Our **detailed** comprehension of ecosystem state shifts in lakes has been possible through ecosystem-scale experiments (1, 2, 7, 10, 11).*

24. Line 19 – suggest inserting “a” - “may also increase a stream’s resistance to stress”

R24. Inserted:

*Line 27: **Indeed**, streams have relatively small instantaneous water volume when compared to lakes and may therefore respond more quickly to environmental changes (**22**) (even though the continuous water renewal may also increase **a** stream’s resistance to stress).*

25. Line 37 – consider replacing “proliferated” with “widespread”

R25. Changed:

*Line 37: **Benthic cyanoHABs** are expected to become more frequent and more impactful due to global warming, **widespread** increases in water use, and nutrient inputs (8, 9, 13).*

26. Line 52 – suggest not deleting “(FR)” as marked, unless you also delete “(FI)” in line 53

R26. Changed:

*Line 51: **The stream analysis consisted of three phases: (1) establishment of baseline conditions (BL) under unaltered flow (~ 1.5 years); (2) experimental diversion of flow, inducing systematic flow reductions (FR) in the downstream reach (~ 0.3 years), and (3) gradual reset to initial flow (FI) conditions in the downstream reach (~ 0.3 years).***

27. Line 58 – suggest replacing “Moreover” with “Additionally”

R27. Changed:

*Line 59: **Additionally**, we compared benthic cyanobacteria biomass between sites upstream and downstream water intakes in seven streams (21 stream-sites) sampled weekly (~2 years: 1456 paired data).*

28. Figure 4 caption – change the estimated breakpoint from “50%” to “58%”? (Last sentence)

R28. We have re-run our analysis with new study sites incorporated in the global literature review and corrected the breakpoint: 58%. See response **R15**. Figure 4 caption states as follows:

Figure 4. Benthic cyanobacteria increase factor according to the relative percentage of flow reduction. a) Measurements (gray circles) of paired cyanobacteria-flow monitoring data ($n = 697$) at the time of the sample in one location upstream of the water intake and two locations downstream water intakes in seven streams from the water supply system; experimental results (blue circles) of cyanobacteria increase with targeted flow reductions including variations $\pm SE$ from temporal replicates within flow reduction (black lines). Rammer-Douglas-Peucker model (RDP) (green line) fitted to monitoring data showing a breakpoint for cyanobacteria increase with a 40% flow reduction. **b)** A global survey of benthic cyanobacteria-flow data ($n = 33$) showing a distribution fitted with RDP model: cyanobacteria increase after a breakpoint of 58% flow reduction (teal blue line).

29. Line 121 – suggest “These temperature and nutrient increases with flow reduction confirm that...”?

R29. Changed as suggested:

Line 124: *These temperature and nutrients increase with flow reduction confirm that mountain streams are sensitive to water abstraction and could abruptly shift to an alternate state when flows are altered (14, 30).*

30. Lines 151-152 – Not sure I understand the added sentence that begins “A longer time flow reduction will elucidate...”. “might elucidate”? I’m unclear what is meant by “as diatoms and green algae recover”; my understanding is that green algae did not respond to the flow manipulation.

R30. This observation is correct and green algae did not respond during the experimental period. We consider the possibility that a longer period of flow reduction might give other responses to understand the nitrate increase and other potential responses to longer flow reduction.

Line 157: *A longer time flow reduction **might elucidate** the underlying mechanism for nitrate increase as we **observed diatoms had a delayed response while green algae showed none.***

31. Line 173 – replace “previously to conduct” with “prior to conducting”

R31. Changed:

Line 178: *We monitored the Chalpi Norte stream for ~1.5 years prior to conducting our experiment for ~0.5 years (176 days), and ~0.4 years after the manipulation.*

32. Line 182 – correct spelling “evaluate”

R32. Edited.

33. Lines 260, 263 – replace “invertebrate’s” with “invertebrate”

R33. Edited.

REVIEWERS' COMMENTS:

Reviewer #1 (Remarks to the Author):

I thank the authors for making the suggested changes. I only now have a few minor suggestions before acceptance of the manuscript.

Line 32. Nodularia

40. references 3, 7, 11 – 7 is not appropriate as benthic. You need a planktonic river reference i.e Mitrovic, S.M., Hardwick, L. and Dorani, F., 2011. Use of flow management to mitigate cyanobacterial blooms in the Lower Darling River, Australia. *Journal of Plankton Research*, 33(2), pp.229-241.

61. downstream of water ...

Figure 4. Im a little confused why there are 4 graphs when I think there should be 2? Are 2 of these the old graphs from last submission and should have been deleted?

125 and nutrient increases with ...

130. global literature survey ...

131 do you mean natural rather than baseline?

136 – clarify what the 10% etc means – i.e. of natural flow etc.

137 – revise the recommendations to what? What suggestion do you have?

140-142. the new line about stagnant shorelines and floating (therefore planktonic?) is not really needed and to me doesn't really fit into this study.

143 quantification of a flow....

144 also allowed identification of the ecosystem

165 combining experimental ...

166 literature analysis this work ...

169 managers should be ...

181 Further, in the nearby

199 fixed percentages using

217 flow and all available